# Participatory action research to address lack of safe water, a community-nominated health priority in rural South Africa

Jennifer Hove[1,2], Denny Mabetha[3], Maria van der Merwe[4], Rhian Twine[1], Kathleen Kahn[1], Sophie Witter[5], Lucia D'Ambruoso[1,2,6,7,8]*

1 MRC/Wits Rural Public Health and Health Transitions Research Unit (Agincourt), School of Public Health, University of the Witwatersrand Johannesburg, Johannesburg, South Africa, 2 Institute of Applied Health Sciences, Aberdeen Centre for Health Data Science, School of Medicine, Medical Sciences and Nutrition, University of Aberdeen, Scotland, United Kingdom, 3 Cochrane South Africa, South African Medical Research Council (MRC), Cape Town, South Africa, 4 Maria van der Merwe Consulting, White River, South Africa, 5 Institute for Global Health and Development, Queen Margaret University, Scotland, United Kingdom, 6 Department of Epidemiology and Global Health, Umeå University, Umeå, Sweden, 7 Public Health, National Health Service (NHS) Grampian, Scotland, United Kingdom, 8 Department of Global Health, Stellenbosch University, Stellenbosch, South Africa

* lucia.dambruoso@abdn.ac.uk

**Data Availability Statement:** The author's study protocol does not allow to share the qualitative data beyond the research team to protect stakeholders

## Abstract

### Background

Despite international evidence supporting community participation in health for improved health outcomes and more responsive and equitable health systems there is little practical evidence on how to do this. This work sought to understand the process involved in collective implementation of a health-related local action plan developed by multiple stakeholders.

### Methods

Communities, government departments and non-government stakeholders convened in three iterative phases of a participatory action research (PAR) learning cycle. Stakeholders were involved in problem identification, development, and implementation of a local action plan, reflection on action, and reiteration of the process. Participants engaged in reflective exercises, exploring how factors such as power and interest impacted success or failure.

### Results

The local action plan was partially successful, with three out of seven action items achieved. High levels of both power and interest were key factors in the achievement of action items. For the achieved items, stakeholders reported that continuous interactions with one another created a shift in both power and interest through ownership of implementation processes. Participants who possessed significant power and influence were able to leverage resources and connections to overcome obstacles and barriers to progress the plan. Lack of financial support, shifting priorities and insufficient buy-in from stakeholders hindered implementation.

and organizations confidentiality. Providing this data publicly would violate the ethics approval received for this study. However, relevant excerpts of the data underlying the results presented here are available from the corresponding author upon reasonable request. Where authors are not available, data may be requested from achds@abdn.ac.uk.

**Funding:** LD acquired funding for this study. This study was funded by the Joint Health Systems Research Initiative from Department for International Development/MRC/Welcome Trust/ Economic and Social Research Council under MR/ N005597/1 and MR/P014844/1. https://wellcome. org/grant-funding/funded-people-and-projects. This work was nested within the MRC/Wits Rural Public Health and Health Transitions Research Unit (Agincourt), supported by the Faculty of Health Sciences, University of then Witwatersrand and the Medical Research Council, South Africa. The funders had no role in study design, data collection and analysis, decision to publish, or preparation of the manuscript.

**Competing interests:** The authors have declared that no competing interests exist.

**Abbreviations:** BBR, Bushbuckridge; COVID-19, Corona Virus Disease of 2019; DoH, Department of Health; DSD, Department of Social Development; EHPS, Environmental Health Practitioners; EHS, Environmental Health Services; HDSS, Health and Demographic Surveillance Site; HIV, Human Immunodeficiency Virus; IUCMA, Inkomati Usuthu Catchment Management Agency; LMIC, Low- and Middle-Income Countries; MDoH, Mpumalanga Department of Health; MHS, Municipal Health Service; MRC, Medical Research Council; NGOs, Non-Governmental Organizations; NHA, National Health Act; NHI, National Health Insurance; PAR, Participatory Action Research; PHC, Primary Health Care; SDGs, Sustainable Development Goals; VAPAR, Verbal Autopsy with Participatory Action Research; WBPHCOT, Ward Based Primary Health Care Outreach Team; WHO, World Health Organization.

## Conclusion

The process offered new ways of thinking and stakeholders were supported to generate local evidence for action and learning. The process also enabled exploration of how different stakeholders with different levels of power and interest coalesce to design, plan, and act on evidence. Creation of safe spaces was achievable, meanwhile changing stakeholders' level of power and interest was possible but challenging. This study suggests that when researchers, service providers and communities are connected as legitimate participants in a learning platform with access to information and decision-making, a shift in power and interest may be feasible.

## Introduction

This paper reports on the process of collective implementation of a health-related local action plan developed in a participatory action research (PAR) process, addressing lack of safe water in rural South Africa. Community participation in primary health care (PHC) gained support in the Declaration of Alma Ata in 1978 by the World Health Organization (WHO) [1]. Normative support for participation has endured. Globally, community participation and development of strong partnerships has been advocated internationally and re-emerged as central in realizing sustainable development goals (SDGs), based on the following concepts: inform, consult, involve, collaborate and empower [2]. Participation progresses a paradigm shift in which communities are viewed not as mere beneficiaries of services, but co-implementors within the development context [2,3].

An important aspect of the participatory approach is an increasing appreciation of the benefits of involving service users in priority setting and collective action [4]. Research evidence worldwide has shown that community participation can lead to improved health outcomes and ensure a more responsive and equitable health system, including, for example, managing the current COVID-19 pandemic [5–12]. Evidence also suggests that participation enhances the legitimacy and acceptability of decisions, furthering trust in public institutions [1,6,8]. Fundamental to PAR is the creation of an enabling environment that addresses issues of representation, capacity, power differentials and sustainability [2,3]. Participation, particularly of vulnerable, marginalized groups in health systems, reflects a rights-based approach to health [13]. In Low- and Middle-Income countries (LMICs) participation is a key element in good governance and community empowerment [14–18].

Several conceptual frameworks and theories have been developed to assess levels of participation, but countries vary in implementation of participation. Arnstein's (1969) ladder of citizen participation is one of the most influential models, which describes participation as eight rungs of a ladder, from non-participation to citizen led control [19]. Arnstein's ladder describes how much influence stakeholders have in the process of decision making and control. When participation is tokenistic, communities have less control over decisions and actions that affect their lives and health. With more complete forms of empowerment, people are in control and have decision-making power. Arnstein stated that "*participation without redistribution of power is an empty and frustrating process for the powerless*" (p.216) [19]. Marginalized communities, however, are often inadequately engaged, under-represented or less empowered to achieve meaningful participation [10,20]. Power is complex, defined as '*capacity or ability to change the probabilities of occurrence of a desired event in a given social context*'

(pg. 8) [21]. In PAR approaches, relationships of power influence the level of participation and outcomes [22].

Since the dawn of democracy in South Africa in 1994, community participation in health has been embedded within policy and guidelines and has evolved over time [23]. Pre-dating PHC Re-engineering, community participation in PHC was institutionalized in the National Health Act of 2003, through establishment of health committees [23]. In addition, community health care teams known as Ward Based Primary Health Care Outreach Teams (WBPHCOTs) were established as part of the PHC re-engineering strategy to deliver services to residents in rural and underserviced areas where access to health care is often limited. However, despite extensive support for community participation, WBPHCOTs and health committees encounter challenges as mechanisms for community participation [24,25]. WBPHCOTS and health committees have not met expectations in terms of representing people's interests, lacking resources, lacking a clear mandate and accountability. Consequently, they are inefficient in addressing local need and amplifying the voices of marginalized communities [24–28]. These structures exist on the periphery of the health system, offering mainly health-related information and providing little opportunity for impact with regards to improved service planning, prioritization and co-production of more relevant evidence in health care [24,27,29].

## Lack of water in Bushbuckridge

Located in the north-eastern region of the Mpumalanga province in South Africa, Bushbuckridge local municipality is a 'presidential nodal point' [30]. Presidential nodal points are typically defined by underdevelopment, contributing minimally to the Gross Domestic Product (GDP), having a high population density, encompassing the poorest rural and urban communities, being structurally disconnected from the developed world and the global economy, and lacking the capacity for self-generated growth [31].

Access to water is a basic human right. South African citizens have a constitutional right to access clean and safe water, and the government has an obligation to ensure that all citizens have access to this basic need [32]. The Bushbuckridge municipality experiences a number of challenges such as high levels of poverty, unemployment, and substantial service delivery backlogs [30,33]. Service delivery backlogs often lead to service delivery protests, as people express their dissatisfaction with the provision of essential municipal services such as water supply, electricity, and sanitation [34,35].

The sources of water in Bushbuckridge Municipality include both surface water and groundwater [32]. The municipality relies on dams, rivers, and boreholes to supply water to the local communities. However, the quality and reliability of the water supply can be affected by various factors such as drought, pollution, and poor infrastructure. In some areas, residents rely on water tankers to deliver water, which can be irregular and insufficient to meet their daily water needs [30]. This situation sometimes results in long queues at water collection points, which can be time-consuming and physically demanding. Furthermore, surface water in Bushbuckridge Municipality is not always safe for consumption due to contamination from various sources, including poor sanitation practices and pollution from agricultural activities. This situation can lead to the spread of water-borne diseases and negatively impact the health of the community.

There are various reasons why Bushbuckridge municipality in Mpumalanga may still lack access to safe water [35]. The situation could be due to a combination of factors, including a shortage of water resources, inadequate infrastructure for water distribution, vandalism, theft and the recent power supply interruptions that impact water pumping [33]. According to the

South African Human Rights Commission (SAHRC) report on water and sanitation, the municipality's water supply infrastructure is outdated and insufficient to meet the growing demand for water in the area [36]. The report highlighted that the municipality's water supply infrastructure is inadequately maintained or upgraded, leading to water leaks, burst pipes, and frequent disruptions to the water supply [36]. In addition, water supply systems may be poorly designed or inefficiently managed, leading to wastage and water losses [35,36].

Responding to the need for practical guidance on community participation programmes in rural LMIC settings, this work aimed to understand how multisectoral action to address a community-nominated priority could be developed and implemented at a local level. This work builds upon previous work on PAR process which we identified health priorities, documented and analyzed the issues, and published our findings elsewhere [37–39]. The objectives were to develop and implement local action plans, to understand how multiple stakeholders influenced the implementation process, and to identify facilitators and barriers to the implementation process in achieving positive outcomes at the local level.

## Methods

### Study setting and design

The work was conducted at the MRC/Wits Rural Public Health and Health Transitions Research Unit (Agincourt) Health and Socio-Demographic Surveillance System (HDSS) in the Bushbuckridge sub-district in Mpumalanga province, northeast South Africa. Established in 1992, the HDSS conducts annual updates, collecting information on vital events including births, deaths, migration, and socioeconomic status to better understand population health in rapidly transitioning societies [40]. The HDSS covers 420km$^2$ with a population of 120,000 people living in 31 villages in 21,500 households, served by two health centers, seven government clinics and three district hospitals.

Infrastructure ranges from fully serviced houses to isolated rural homesteads accessing water from communal standpipes [40]. Bushbuckridge local municipality faces several challenges. These include high poverty levels attributed to high unemployment, limited economic base that results in extensive labor migration. While the place is a well-known tourist attraction, there are pervasive service delivery backlogs (water shortages, sanitation, and roads) and lack of large scale water supply infrastructure, reservoirs, and a reticulation system [35,40–42]. In 2001, the municipality was declared a presidential nodal point [42]. With poverty widespread, many households rely on government social grants as a key source of income [40]. HIV prevalence is high, and the rates of non-communicable diseases are steadily increasing among the population [40,41].

This study draws data from a programme of research titled Verbal Autopsy (VA) and Participatory Action Research (PAR) programme (VAPAR) (www.vapar.org) [43]. VAPAR is a collaborative learning platform established in the rural province of Mpumalanga to develop a partnership of local and international researchers, community members and authorities to co-produce and act on relevant local evidence [38,44]. Verbal Autopsy (VA) is a method used to determine the cause of deaths in settings where deaths are not medically certified [45,46]. It involves interviewing a family member of the deceased or a caregiver to collect information about the symptoms and circumstances leading to death [45,46]. PAR is a collaborative and democratic research approach that involves active participation of stakeholders (people affected) in the research process to address social issues and bring about positive change [47–50].

The VAPAR programme was designed to progress multi-sectoral engagement at different levels while prioritizing the needs of the communities to better understand their health problems and developing collective actions [37,38]. Lack of safe water was nominated as a priority

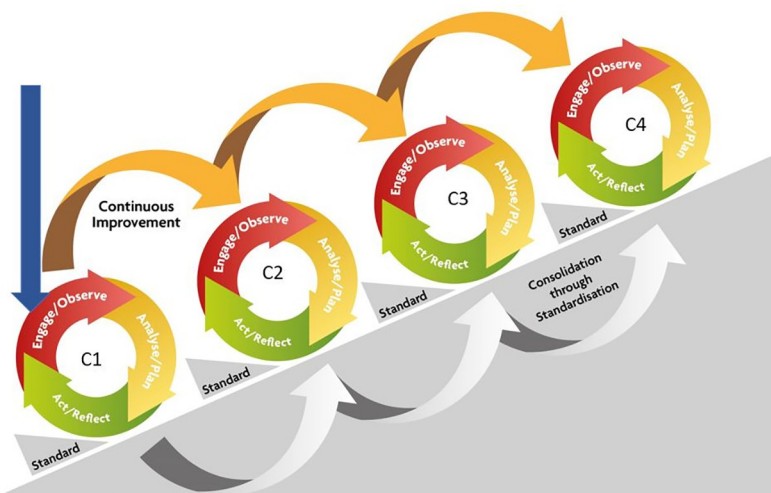

**Fig 1. VAPAR schema developed by the research team illustrating participatory action cycles to tackle the issue of lack of safe water, which was identified as a top health priority by the community.**

health topic in one of the villages in Agincourt HDSS [37–39]. The VAPAR programme consists of a series of iterative five action learning cycles following a stepwise approach [43,44,51]. In this paper we report on the first cycle of the five, which comprised three steps described below and as shown in Fig 1. Cycle one involved a reiteration and adoption of the process in order to transfer ownership and embed it into the local context, incrementally and progressively. These steps in cycle one were crucial in informing the subsequent cycles reported elsewhere [52].

In this process, the research team and participants worked collaboratively in all stages of the research process, including problem identification, data collection, analysis, interpretation as well as the implementation of the local action items developed to address lack of safe water. The research team's role was to facilitate and support the participation of both community and government stakeholders in the research process, providing technical expertise, prepare workshop guides and ensuring that the research was rigorous and met ethical standards. The research team actively listened to and incorporated the perspectives and needs of the community stakeholders. The participants' role as partners was to actively engage in the process, share their knowledge and experience, and contribute to problem-solving and decision making. Overall, the level participation was high for both the research team and participants, playing active and equal role in the research process.

## Step 1: Engage and observe

VAPAR staff re-engaged 24 community stakeholders who had been involved in the VAPAR pilot work in 2015 to maintain and sustain relationships [53]. Three villages were purposively selected based on varied accessibility to health care facilities and socio-economic status. Community members within each of the three villages were purposively sampled to understand the subjective reality and experiences using maximum variation sampling. Participants represented a cross-section of the community and included community and religious leaders, community health workers, traditional healers, and family members. In the first workshop,63% of the participants were females.

In each village, an initial workshop was held with eight original participants, where the study was introduced using PAR methods. Through facilitated discussions, participants listed a range of priority health topics. Participants used participatory data collection tools such as ranking and voting, a systematic way to collectively narrow down number of topics, into one topic that represents the collective perception of the group. One village nominated lack of safe water as a priority health issue, while in the other two villages, alcohol, and other drug abuse (AOD) was nominated as a priority health issue. This paper reports on lack of safe water in one village, AOD abuse findings are presented elsewhere [39].

In order to expand the participant base, original participants were asked to identify an additional eight participants from the same village. This approach aimed to bring in knowledgeable voices from those typically excluded and directly affected by lack of safe water, thereby supporting the sharing of ownership and control over the process. Women of reproductive age (18–50 years in this study) were identified by participants and recruited by the researchers. The village-based discussion group was then held with 16 participants, including original and new participants, and was convened in a series of seven workshops held in August-September 2017 (Fig 2). Participants were involved in co-designing the process and co-production of evidence through identifying the priority health topic, identifying new additional participants, co-facilitating, and through practical aspects such as choosing times and venues for workshops.

PAR tools were used to systemize subjective experience to develop collective accounts, to identify cause and effect relationships at various levels, to build shared understanding of the relationship between key actors and to explore priorities for action to address lack of safe water. The findings were collectively validated based on insights and perspectives of the group with outputs recorded and appraised. Photovoice was used as an additional input to the process a means of empowering participants to share their experience and perspective in a visual and accessible way. Participants were provided with cameras and asked to take photos of their daily lives and experiences related to the research topic. The photos were used as a basis for group discussions and analysis, allowing participants to share their insight and perspectives in a way that was visual and participatory. Workshops were held in xiTsonga, the commonly spoken language in the area, and discussions were audio recorded, transcribed, and translated for

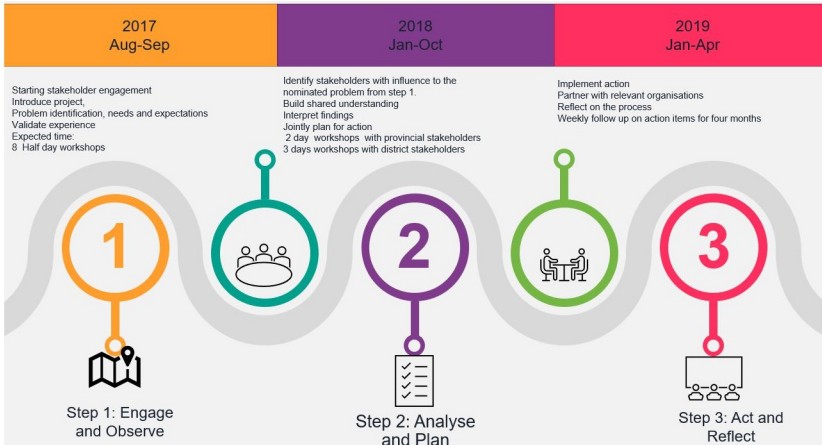

**Fig 2. Stakeholder engagement timeline: Activities were consistently communicated throughout the project and stakeholders were kept informed about upcoming meetings, workshops and were involved in planning the next steps, including creating meeting agendas and remained informed throughout the project.**

narrative analysis. In addition, flip charts were used to display a collective account. Results from this stage were organized into a research brief for the next step. The process is explained further elsewhere [38].

## Step 2: Analyze and plan

We then held a series of facilitated workshops from January to April 2018 with government and community stakeholders to analyze and plan based on the research brief and PAR data from the Step1. The first workshop convened 13 officials from Mpumalanga Department of Health (MDoH) (district and province level) in the city of Mbombela in Ehlanzeni district, Mpumalanga province. There was recognition that the problem of lack of safe water lies outside the health system and requires multi-sectoral representation. As part of co-design, participants decided which sectors needed to be invited to be involved in the process. Therefore, the next two workshops brought together various role players from the community, and representatives from provincial and district MDoH, department of water and sanitation (DWS), department of basic education (DOBE), Department social Development (DSD), Cooperative Governance and Traditional Affairs (COGTA), and Culture, Sports, and Recreation. There were 12 and 15 stakeholders in the second and third workshops respectively. Outputs from the first workshop were discussed in workshop two, using rich pictures and group model building to develop consensus and shared understanding of lack of water, implications on health, challenges, potential solutions and on policy and strategy contexts. In the third workshop, stakeholders collectively developed recommendations for action with a focus on local implementation contexts. To create a more inclusive and equitable research process, participatory tools and techniques used were adapted to the level of capacity of all stakeholders to avoid any form of exclusion.

Two further interactive planning workshops were then held to develop inter-sectoral partnerships and action plans from September-October 2018 at the Agincourt HDSS offices in Bushbuckridge sub-district (Fig 2). In workshop four, we re-defined the focus topic to be 'lack of safe water and its effect on child health' to reflect and accommodate priorities of both community and DoH stakeholders. In workshop five, through facilitated discussion in groups, we explored and determined what action should take place, by whom, when and with what resources (Fig 3). We then developed a local action plan with seven action items through consensus and feasibility appraisal with consideration of current policies and existing strategies and action that could be taken by local stakeholders. All the stakeholders pledged their commitment to execute the action plans.

## Step 3: Act/Reflect

We conducted a series of follow-up visits to stakeholders who had committed to leading the implementation of the action items during January–April 2019. During the follow up visits we appraised actions, actors, and the timing of the action, and revisited the baseline and targets set in the co-designed local action plan. A structured 'record of action' was used to capture reflections from participants who committed to lead, on mechanisms through which changes occurred during implementation, if at all; on impacts and whether and how these addressed health equity and empowerment; whether and how these related to the use of evidence; acceptability of the process; and how opportunities for interactions and reflections between participants were created.

At the end of four months, 27 follow-up visits had been completed and an additional eight follow-ups done telephonically with participants from the community, government departments and NGOs at sub-district and district levels who had committed to progress the action

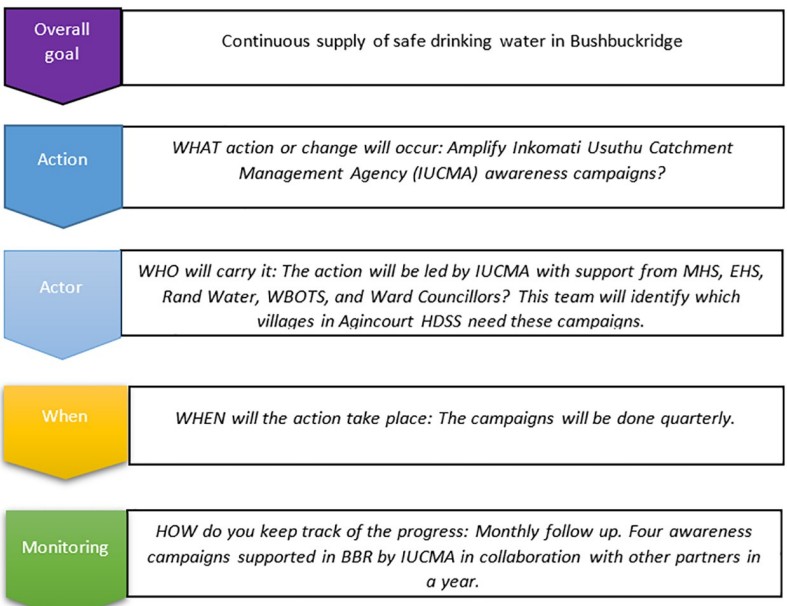

**Fig 3. An example of how an action plan was developed, addressing lack of safe water.**

items. Participants were involved in more than one action item, which minimized the number of visits since two or three action items could be monitored in one visit. During follow up visits, participants reported on progress of the action plan, for example how the action was started, interim steps, transparency and information sharing, meetings held, or coordination with other departments. The research team convened weekly to discuss initial impressions and reflections from the participants' perspectives and experience. Discussion and information reported was recorded on the record of action and signed by both the stakeholder leading the action item and the study coordinator. At the end of the implementation period, we developed categories of achievement for each action item, accompanied by detailed narratives and reflections. The findings were shared with all the stakeholders by the researchers.

### Data analysis and management: Mendelow's matrix

We used Mendelow's matrix (1991) to understand the process involved in collective implementation of a health related local action plan with community stakeholders [54]. Mendelow (1991) classifies stakeholders on a grid whose axes are 'power' to influence and 'interest', Fig 4 [54]. Power is the influence, ability or capacity created by the relationship between resources and interest [55]. This approach to stakeholder analysis allowed us to explore power dynamics, and how power was shared to shift low power participants to high power [56]. Data were drawn from: follow up visits' formal and informal feedback, presentations, meeting minutes, completed records of action, observational notes, workshop reports, visual images, and the study coordinator's reflective journal. Data were stored in Microsoft Word and image files. Data were managed by researchers and stored on secure university servers.

The research team used NVIVO, a data management software, to analyze the data and synthesize information from the participants. This enabled the identification of facilitators and barriers to the implementation process. Going through the record of action, notes and reflective journals, the research team verified the evidence from the participants with regards to

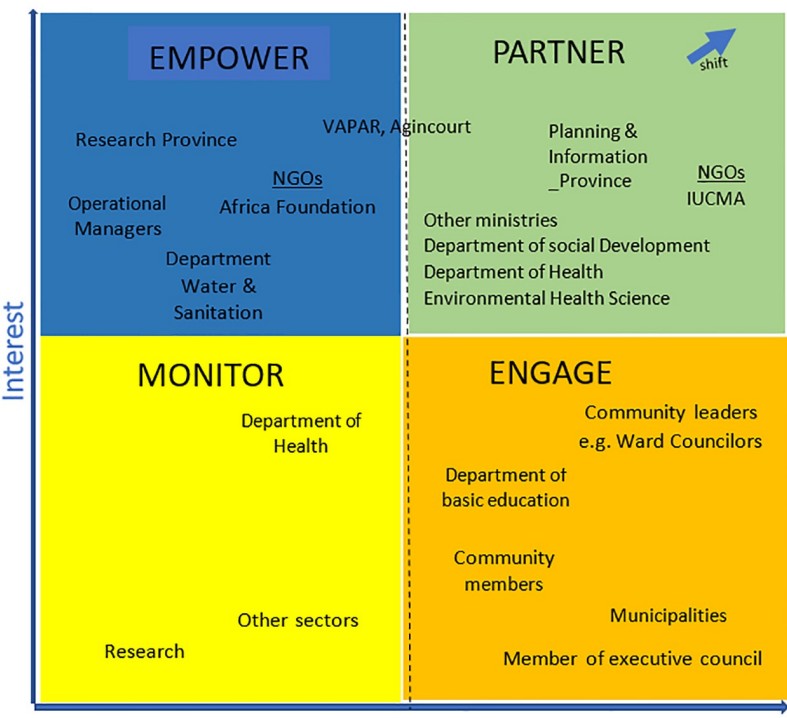

**Fig 4. Stakeholder analysis using Mendelow's power grid, with arrow on the far right indicating overall shift in power and interest of stakeholders.**

whether action was achieved or not. The participants were consistently engaged telephonically throughout the analysis phase to minimize the impact of individual biases or interpretations. Furthermore, the research team prioritized the interpretations and perspectives of the participants in analyzing the impact of the study.

## Ethical considerations

Ethical considerations related to PAR were observed from the beginning. The role of the researchers, democratic and power sharing principles, ownership, and control of the research were discussed without manipulation, being aware of our role, communicating openness to debate, equality of voices and subjective reality. Prior to their participation in the research, written consent was sought from all participants who were 18 years or older, explaining the details of the research at a level understandable to everyone so that participants could make an informed decision. To protect the privacy, safety of participant's personal information, and prevent any potential harm or negative consequences, codes were assigned in place of their real names, data was stored securely and confidentially. Only authorized individuals had access to the data. Participants were provided with refreshments and transport costs and were reimbursed for time spent in workshops with 300ZAR per participant at each step. The research was approved by Institutional Review Boards at the Universities of the Witwatersrand (M1704155; M190222) in South Africa and of Aberdeen (CERB/2017/14/1457; CERB2017/9/1518) in the UK and the Department of Health and Social Development (MP_201712_003) in Mpumalanga.

## Results

The results are presented under two main sections: outputs from the implementation process of the local action plan, and reflections from participants. In Table 1, we summarize progress made towards action plans.

### Achieved action items

Out of the seven local action items planned, three were achieved. These were (1) *river clean-up campaigns*, (2) *awareness campaigns* and (3) *building trust between communities and municipalities*. Follow-up visit reports on achieved action plans highlighted high interest and commitment of the participants in holding key meetings as planned. In *river clean up and awareness campaigns* led by Inkomati Usuthu Catchment Management Agency (IUCMA), regular dialogue and engagement of various stakeholders were reported. Multi-stakeholder engagement was reported to be instrumental in the establishment of relationships, peer learning, trust and collective decision making. The researchers and the wider community joined one of the river clean-up campaigns as shown in Fig 5.

IUCMA's mandate is to enable everyone to participate in planning and making decisions regarding water related issues that affect their life [57]. Various stakeholders, including researchers and community representatives, were engaged in an *IUCMA awareness campaign* workshop. The purpose of the IUCMA awareness campaign workshop was to train and provide technical assistance with the intention to empower communities with knowledge and information needed for taking part in water resources management and making informed decisions. IUCMA had power and were highly interested, hence they provided the leadership and resources needed for implementation of the river clean up and awareness campaigns. These two action items were embedded within the IUCMA existing infrastructure.

> *"Attending these IUCMA workshops were important to me, made me understand the value of working together, now I understand that it is my responsibility to keep my environment clean, I no longer rely on government do everything for me. I will mobilize others in my community to come to the next workshop and learn"*
>
> *(Community stakeholder)*

The third action achieved was to *build trust* between communities and the municipality. Community stakeholders reported dissatisfaction and mistrust with water authorities and the municipalities regarding water service delivery. Implementation of this action depended on optimizing positive contact between the municipality and community members, to bring municipality closer to the communities. Community stakeholders were highly interested in creating dialogue with municipalities, and researchers facilitated meetings with the two local municipalities, as well as a community workshop. Municipalities, through sub-regional managers connected with the community to respond to water challenges communities' encounter. It was reported in the meeting that the community organized with the municipality that water is a challenge in Bushbuckridge. The eastern part of Bushbuckridge relies on a borehole system for its water supply. It was highlighted that boreholes have a short life span due to technical problems and pipes that always burst, which makes them a financial burden. Regular community meetings with the municipality were recommended as an opportunity for the community to air concerns, understand how municipalities work and find ways to address water challenges. Action was based on pragmatic knowledge of the situation with minimum resources required and leadership and facilitation was enhanced by the researchers.

**Table 1. Progress towards the action plan implemented in Agincourt Demographic Health Surveillance Site Villages.**

| Action Item | Baseline | Implementation mechanisms | Outcome |
|---|---|---|---|
| 1. River clean-up campaign | Individual entities have clean-up campaigns, but other stakeholders are not involved | This action was led by IUCMA in partnership with other organizations. IUCMA coordinated with various stakeholders including communities in the study site. Community representatives including the research team also joined IUCMA stakeholder engagement team | Achieved |
| 2. Amplify CMA awareness campaigns. | Four awareness campaigns yearly in BBR supported by IUCMA. Event has been happening but does not include other stakeholders for example Department of Health, Department Social Development. | Stakeholders (IUCMA) leading the action plan reported overwhelming support from the broader community and high attendance reported for the first quarter. The researchers and community members in Agincourt HDSS joined the IUCMA awareness campaign workshop held in Dumphries C village in the study site. Workshop participants were informed on water use licensing, water quantity and quality and compliance, illegal sand mining, and conservation (protection of wetlands and topsoil). | Achieved |
| 3. Build trust between Municipality and community | No dialogue. Activism campaign. Communities coming together: "LET'S TALK ABOUT IT" to enable political tolerance | The research team created dialogue between communities and municipality. Contact was made with Agincourt and Lillydale Municipal office in Agincourt HDSS to build relationships and create partnership. Municipalities (two) in the study site agreed to work with VAPAR to understand community needs. An official from Lillydale municipality attended one of the workshops to respond to community issues. Through this we managed to create dialogue between communities and municipalities. The municipality reported that plans were underway to change to a bulky water system from Inyaka dam | Achieved |
| 4. Advocate for a recycling center | No recycling center in the study site. | This was a long-term action hence the initial steps were initiated, and consultation made with relevant stakeholders. Africa Foundation led this action. Follow up visits highlighted that there is a place to establish the center. Africa Foundation is an NGO in BBR working in partnership with Sabi Sands reserve, which support it financially. Africa foundation had the land and resources for a center in Lilydale to cater for the villages in Agincourt HDSS. | Partially Achieved |
| 5. Water harvesting to attention of school and clinics. | Nothing was happening during the time of our process in Sep 2019 Some schools we're harvesting water but infrastructure not maintained and had stopped. | Led by DoE via Schools and DoH via clinic. Stakeholders leading the process reported challenges during implementation. School governing bodies were meant to provide budget support and DoH had no budget. Resources which were donated to school were not maintained by the municipality. The researchers met with Agincourt and Lilydale municipal managers to find out more about water service delivery at schools and clinics. BBR is faced with water challenges and the municipality reported struggling to maintain some of the borehole system the communities are using. | Not Achieved |
| 6. Advocate for water testing partnerships | Some water testing currently being done by DoH EHSs at some clinics. In the communities Africa Foundation test water for quality. | Africa foundation was nominated to lead they currently test borehole water. Africa Foundation advocated for collaboration between private/NGO partners to tests borehole and water tankers with support from Rand Water, municipality. Challenges were reported on clarity on roles and responsibilities on testing water. Africa Foundation only test borehole water that they control. Africa foundation tried to bring municipalities on board but all the boreholes they have assigned to municipalities were not maintained. | Not achieved |
| 7. Orient Executive Mayor to VAPAR | Mayor not familiar with the VAPAR research work | The researchers found it hard to engage with the mayor. The mayor's office was busy preparing 2019 elections. The mayor confirmed availability in May 2019 after so many failed attempts from the beginning of the implementation period of Jan 2019-April 2018. | Not achieved |

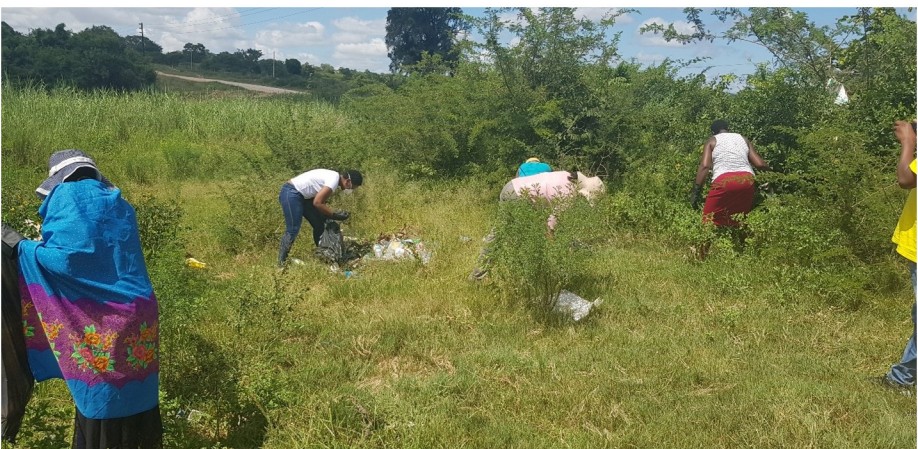

**Fig 5. River clean-up campaign, community and researchers picking up litter along the river in Bushbuckridge district, process led by UCMA (image reproduced with permission).**

> *"I have learnt that some problems don't need a lot of money to be solved but need peaceful communication between people who are affected and concerned with the problem. The idea is to understand the problem first before attempting to solve it".*

> *(Community stakeholder)*

## Partially achieved action items

One action item, *advocating for a recycling centre*, was partially achieved. Africa Foundation, an NGO, committed to lead this action item, as its mandate is facilitating the fulfilment of needs identified by communities, communicating those needs to potential donors, and working with community leaders and project champions [58]. Africa Foundation had the land available to establish a recycling center and was already in partnership with other organizations with strong financial support. Initial contacts were made with relevant stakeholders, and resources were available, but the action was partially achieved as this was a long-term plan and could not happen within the four-month implementation period.

Despite limited time, follow-up visits revealed that the stakeholders were interested and also had power to influence implementation of the action as they had resources allocated for the action. The land was reserved for the recycling center but there was a lack of coordination among stakeholders, and the action was left to the sole responsibility of Africa Foundation. Some stakeholders were not actively engaged and did not share the responsibility for the implementation, as per original planning, owing to multiple competing priorities among government sector stakeholders.

## Not achieved action items

Follow up reports indicated a sizeable gap between planned activities and what was executed for the three action items. The actions items not achieved were (1) *water harvesting, (2) water testing* and (3) *orienting the mayor to VAPAR programme*, Table 1. DBE and the DoH at sub-district level participants volunteered to progress bringing *water harvesting* to the attention of schools and clinics while also enhancing individual skills and social participation and inclusion. DBE stakeholders who had committed to progress the action could not adhere to the

planned activities due to lack of financial support and shifting priorities, despite policy support for rainwater harvesting at schools and clinics. In addition, DoH stakeholders at provincial level, though they had power, did not attend or facilitate any meeting, nor dedicate any resources to the implementation of this action due to work commitments and other competing priorities. There were no budgets from school or clinics to maintain donated infrastructure at the schools and clinics.

Africa Foundation was also nominated to lead the action item on *water testing*. The Africa Foundation had the power to influence implementation of this action item as they already were testing borehole water. They were interested, hence advocated for collaboration between NGOs and the private sector, with support from the municipality. Data analysis revealed that no meetings with stakeholders were held as planned. However, the follow up visits revealed that the Africa Foundation attempted to invite stakeholders for planning meetings to execute the action plan, but stakeholders were not available. Though targeted stakeholders' municipality and local run NGOs showed interest during the planning process, stakeholders reported multiple competing priorities, and lack of incentives. This action depended on departmental funding, hence there was lack of ability to mobilize resources for implementation, hence no dialogue was facilitated. There was increased awareness by committed stakeholders of the complexity of bringing multi-level actors together during the implementation process than had been previously recognized during planning.

The research team was assigned to *orient and share information about the project with the Bushbuckridge Executive Mayor office*. This was critical to bring the mayor on board to ensure political buy-in and to navigate the complex political environment. Several attempts were made to engage the mayor by the research team at the beginning of the implementation period from January to April 2019. The mayor was not available due to the timing as the mayor's office was preoccupied with preparations for the 2019 municipal elections. The office of the mayor had multiple competing priorities, and more attention was paid to political duties.

## Reflections

**Community stakeholders.**   The process was reported as valuable and significant as it created an opportunity for communities adversely affected by lack of water to connect and constructively and actively engage with authorities/ service providers in appropriate and accessible venues. For the achieved action items, stakeholders reported that continuous interactions with one another created a shift in both power and interest through ownership of the implementation process. Community discussions to engage and bring together many voices of the community created spaces and opportunities to expand and connect with stakeholders outside their communities. Dialogue between communities and government stakeholders helped to bring positive change, as community members began to recognize their own expertise, and this contributed to improved interest, commitment and buy in of multiple stakeholders. The process facilitated peer to peer learning and exchange of knowledge. Local community members reported improved awareness and understanding of the various stakeholders within the water supply chain, and the bureaucratic procedures among government departments. For example, a lack of water in the taps does not always imply a challenge at sub-district municipal level.

> *"Yes, because when we build good relationships with our service providers it is easy for them to attend to our issues when we need them. Like if today the municipality manager had come, and we could tell him peacefully what our problems are even if he doesn't have all the answers today, he may be able to consider or address them in future"*

*(Community stakeholder)*

Community members reported having a deep sense of agency and belonging and felt that their voice was heard and included in decision making. Stakeholders were treated as equals; decisions were negotiated, and clearly communicated in a transparent way to ensure sharing of power. Community stakeholders reported that coordinating and active interaction during implementation resulted in improved confidence, communication, and problem-solving skills as they recognized that their voice matters.

*"I think for me participating in VAPAR improved my abilities to identify whatever problem I might have, sit down with other people, and relevant stakeholders and talk as a way of trying to find solutions than to just go on strike and destroy the little that the government has provided to us"*

*(Community stakeholder)*

*"Here we are all equal, there is no one who is smarter than the other. VAPAR team allows everyone to speak their minds, we are not afraid to raise our issues. They also allow as to be part of the decision-making process in everything that we are doing here. This makes us feel in control the process"*

*(community stakeholder)*

Stakeholders made a commitment that in future, if there are service delivery challenges related to water, they would resort to dialogue instead of protests, which are often violent and destructive. Community members reported increased interest in IUCMA's river clean up and awareness campaigns as they participated in the events and promised to join in future events.

*"There have been a lot of service delivery protests in communities, but they did not accomplish much; everyone realized that it is time to shift our ways of thinking and initiate dialogue, unite and collaborate and create sustainable partnerships to solve community problems"*

*(Community stakeholder)*

**Government stakeholders.** Attitudes and perceptions of government stakeholders shifted as they began to value and understand community voice. Overall, the deliberative process provided opportunities for shared learning, prompted a shift in power relations and collective responsibility. Stakeholders appreciated the process and its iterative nature, trying to systematically solve a problem gradually, making small changes over time in a subtle way. The process promoted integration of a variety of actors in collective action and decision making. In addition, the municipal managers encouraged community members to come forth with relevant information, and to report vandalism and illegal water connections.

*"This programme was important for me. I got insight on how other government sectors operate and working together with communities and other sectors was empowering. I also learnt how to engage service users in decision making and planning"*

*(Government stakeholder)*

Participants reported the process as instrumental in ensuring that interventions are aligned to local priorities. The process created safe platforms for dialogue for other government departments, for example DoH and DSD which reported that they do not have an active role in water governance.

**Researcher reflections.** The research team was composed of both local and international researchers, consisting of individuals from various backgrounds, including those who identify as black and white. During our initial fieldwork interactions, participants directed their communication in Xitsonga solely towards the black researchers, as the population in the study area was exclusively black African. Nevertheless, the project's local coordinator provided support in instances where the Xitsonga language proficiency of the research team was limited. Building relationships with the participants was crucial to addressing power dynamics, which are a fundamental element of PAR.

At the outset of the research process, the research team possessed greater power due to their access to resources and involvement in the initial design and planning stages. This included the development of topic guides for workshops and recruitment of the first cohort of participants. To mitigate potential power imbalances, the team engaged in reflexive discussions during the workshops and co-developed local action plans and joint implementation. However, the researchers were able to reflect on their positionality and gain a deeper understanding of the needs and perspectives of the participants.

The research team did not anticipate that lack of access to safe water would emerge as a health priority topic identified by community stakeholders, even though water is a key social determinant of health. At this juncture, there was a shift in power dynamics, and the role of the community stakeholders evolved from being mere informants to active agents in the data collection, action planning, and implementation phases. It has been appreciated that communities possess a nuanced understanding of their problems, and their insights are invaluable to informing effective solutions. Through engagement with the community, it was discovered that households desired a tap in each home. Regrettably, due to limited time and resources, this objective could not be achieved by the end of the first cycle examined in this paper. Nevertheless, the PAR approach facilitated the creation of neutral and safe spaces for dialogue between the community and service providers, which was a significant step in fostering constructive engagement and addressing the identified water crisis.

Collaboration with government stakeholders during the research process provided invaluable insight into the complexities and bureaucratic procedures involved in implementing policies across different government departments. We also encountered challenges in bridging the gap between knowledge and action as only three out of the seven local action items were achieved. The successful implementation of certain local action items hinged on the cooperation of local government officials and key stakeholders. However, the lack of financial support, shifting priorities and buy-in from these parties hindered the implementation of the local action plan as originally intended.

Despite these challenges, repeated interactions and collective planning and implementation enabled us to re-establish trust and forge new connections with sub-regional Bushbuckridge local municipalities. As a result of these efforts, the community stakeholders and municipalities made a joint commitment to tackle the issue of vandalism of water infrastructure, as well as corruption, to address the pressing water challenges in the area. By ceding power and control to stakeholders, unanticipated actions were able to emerge 'out of cycle' leading to the development of an organic process to emerge that was responsive to stakeholders needs and priorities, and reflective of the participatory nature of the process. Regardless of some local action items not being achieved, the reiteration of the process; reflection on action and co-design was key element prior to the subsequent cycles. This involved revisiting the research questions and identifying new stakeholders who could support the implementation process going forward. This collaborative approach fostered meaningful connections between diverse stakeholders. Stakeholders were able to gain a deeper understanding of how power and interest influence programmes and how they can be harnessed to drive success.

During the planning phase, it was crucial to time the project in a way that ensured the mayor's availability for support, buy-in, and participation, without other commitments. This is particularly important during election periods when the mayor's availability may be limited.

## Discussion

The arrangements of the findings by the power/ interest analytical framework offers insights on how power and interest influenced implementation of local action plans. This process enabled exploration of how different stakeholders with different levels of power and interest coalesce to design, plan, and act on evidence. The three action items achieved had action leads with high interests and high power. The stakeholders leading the action items were not working in silos, but consensus decision making was facilitated regardless of level of power and interest, and they convened with all stakeholders as planned. The organizations and departments leading the actions offered resources and leadership with inputs from all stakeholders, they coalesced around commitment to action, and that had the potential to reduce power imbalances creating a culture of collective learning, and action.

This study suggests that when researchers, service providers and communities are connected as legitimate participants in a platform that provides equal learning opportunities, access to information and decision making, a shift in power and interest may be more feasible. Stakeholders leading the action items created conditions that enabled the other stakeholders to be actively involved to achieve collective action and an opportunity to learn by doing. When people work as a collective, a shared power and a shared interest can become feasible [59–62]. Continuous interactive engagement of all stakeholders, those with both low and high power and interest, allowed different voices, building, and strengthening capacities and decision-making power.

The implementation process of the action items created spaces for collaborative dialogue while power was negotiated among stakeholders. This was observed by understanding different backgrounds of participants, revisiting, renegotiating the PAR principles and putting those into action in collective advancement of the local action plan. The findings resonate with other research evidence which suggests that continuous collaboration of multiple stakeholders has potential to strengthen capacities of communities through development of skills, knowledge, and shared learning [63–66]. Empowerment of stakeholders that arise through continuous interaction and decision making in progressing action has the potential to shift power imbalances among the researchers and stakeholders [18,67–69]. Furthermore, implementation process that allow for collective learning increases the legitimacy of the process and sense of agency in previously marginalized, low interest and low power community stakeholders [22,59,70]. As reflected in our study and the wider literature, power and interest are fluid, and have the potential to change [56,63,71]. Egid at al [63], concluded that power is pervasive, hence tools, reflections and actions must provide opportunities to address power imbalances in participatory action research so as to move away from tokenistic co-design.

Consistent with other studies, our findings also show that in the action items partially and not achieved factors that influence implementation are complex [7,22,72–74]. Though challenging, continuous multi-sectoral stakeholder collaboration between community and government/ NGOs actors is essential to ensure that plans are turned into action [4,66,75–77]. Financial support is crucial for sustaining programmes, but it is important to also establish mechanisms for ongoing collaboration and decision making to prevent shifting priorities. This includes regular stakeholder meetings and reviews of progress towards goals, as well as establishing clear communication channels. In addition, power and interest is invariably shaped by the ability of the stakeholders to mobilize resources and their willingness to initiate and lead

the action [55,63]. DoH and DoE stakeholders lacked the ability to mobilize resources needed to execute the action items, and underestimated the time required, hence reported being overwhelmed, despite initial willingness to lead the action. Salka et al [78] concluded that lack of financial resources for providing services potentially hinders effective implementation of set goals and policies. Most scholars agree that high power and high interest coupled with resources has the potential to influence decision and implementation processes [63,68,78]. Resources include time, expertise, information, money and even political support where necessary [78].

High interest was linked with knowledge, information, commitment, and capacity to facilitate implementation of the local action plan. These factors are interrelated. For example, high interest (commitment, information, and capacity) without adequate power to mobilize resources is a major barrier to implementation. Our results highlight that all these factors are critical for the implementation of action plans. Hauck et al [79] conducted a case study on PAR in natural resources management. The authors found that the implementation of local action plans faced challenges such as lack of resources, power imbalance among stakeholders, shifting priorities and limited government support [79].

Creating safe space for dialogue was feasible and achievable. Reflections on the process reported the power of safe spaces for dialogue, strengthened capacity through regular interaction with others and generating and acting on community evidence on community nominated priorities. However, changing the stakeholders' level of power and interest was more challenging. For example, IUCMA was able to advance the action, but rural community dwellers had less ability to do so. In future application of this methods, careful attention to power and interest of stakeholders need consideration during implementation [22,67,80,81]. Power dynamics in multi-stakeholder collaborations if not careful monitored and managed, result in more powerful stakeholders dominating and subtly influencing low powerful stakeholders to their advantage [22,59]. However, establishing shared ownership and responsibility for the initiative can help ensure that all stakeholders remain committed to its success and that priorities remain aligned.

A gap in ownership was evident in the execution of some action items. It was observed in the not achieved action items that some of the stakeholders' interest fell away as their organization/ department priorities diverted them. Community stakeholders were strongly supportive of the implementation process (high interest), but they did not have power in terms of resources (financial/time) to mobilize action. Perhaps unrealistic plans, as well as low power, hindered the ability to implement some of the local action items, as some stakeholders lacked understanding of what they could achieve at a local level with restricted resources.

The wider literature offers important insights. In a case study on community led maternal health programs in rural Bangladesh, scaling up community mobilization was successfully initiated without financial interventions or increase human resource [70]. However, the case study highlighted the challenges encountered during implementation of the program, such as limited resources, low levels of literacy among some community members and a lack of support from some government officials [70]. These challenges underscore the importance of addressing structural and systemic issues that can impact the success of community-based interventions [70].

## Methodological reflections

The Agincourt HDSS, a stable public health observatory, enabled the research team to create safe spaces connecting otherwise disconnected actors in an action learning process across a range of government departments. Despite action plans being partially achieved, the process

was deemed to be positive and useful as it was conducted in the very communities where change had to happen. A deliberative bottom-up approach, engaging stakeholders right from priority setting through design, planning and implementation enhanced the legitimacy of the process. Co-designing, and co-production of knowledge and reflections on action between researchers, communities and service providers promoted a sense of ownership, and community stakeholders felt empowered. However, throughout the process, some government stakeholders were unable to attend all meetings and workshops and had to send representatives in their place, attending again when they were available. This disrupted the flow of the implementation process as certain aspects of the process had to be re-explained to the new participants who were not present in previous discussions.

Stakeholders acknowledged that they were able to understand the process, and that learning from other stakeholders was key in transforming and shifting mindsets, building confidence, and empowering ownership. Power was derived from the information and expertise that stakeholders acquired during the collective implementation of the local action. Evidence produced through the PAR process was relevant and context specific. When we came to collectively reflect on and adapt the process for the subsequent cycle, health system officials recommended integration into routine health system planning to support community participation in PHC [51]. This reflects the epistemological fundamentals of PAR for democratic knowledge production that addresses local needs and enables its translation into action in the form of policy, practice or behavior change in individuals or institutions [48,49,71,82,83]. Co-design and reiteration enabled a more specific and embedded process that was more deeply owned and controlled by stakeholders.

Stakeholders developed an increased sense of ownership by identifying their own resources to solve community problems. The process required time, commitment and resources to build and sustain relationships and could not be covered as an additional item on the already full agendas of some stakeholders in full-time employment [63,84]. As this study only analyzed implementation of the local action plan at the end of the sixteen months cycle, the implementation period was limited. At the end of the cycle, we collectively reflected and adapted to improve context specificity and for practical relevance. Through this process, health system officials recommended its integration into routine statutory planning [51].

## Conclusion

This paper examined processes of implementation of a local action plan through multi-sectoral and community-based collaboration. The process provided opportunities for community stakeholders to connect with service providers and officials in spaces that enabled collective learning, co-production of local evidence and creation of partnerships. While the local action plan was only partially achieved, the process provided a nuanced understanding of implementation processes from the experience and perspectives of stakeholders leading the action plans. The collective decision-making process enabled some stakeholders to shift their power and interests towards addressing social determinants of health. Through the collective, interest could increase if you already had power and power could increase if one already has interest. This process, though contextually bound, offers insights into effective ways to engage multisectoral stakeholders and could guide other HDSSs in the region. It is important to acknowledge that some local action items require longer implementation time than originally anticipated, and setbacks and challenges are a natural part of the process.

However, more attention to power relations during planning and design of multisectoral collaborations processes need consideration to address social injustices. The lack of dedicated resources (time/finance), and institutional support hindered overall implementation of the

local action plan. By understanding these challenges, researchers and practitioners can better design and implement participatory research programmes that can effectively create meaningful change in communities. Multi-sectoral collaborations can be effective in addressing social determinants of health, but adequate resources and institutional support are essential, and attention to power dynamics is necessary to ensure equitable outcomes.

## Acknowledgments

The authors would like to thank all community stakeholders, government, and NGO stakeholders study participants for agreeing to be part of the process, and for sharing their time, knowledge, and perspectives. Thanks also to the Verbal Autopsy with Participatory Action Research (VAPAR) team and staff of the Medical Research Council (MRC)/Wits Rural Public Health and Health Transitions Research Unit (Agincourt), especially Simon Khoza, Khanyisa Ngobeni and Palesa Mataboge. Permissions have been secured for the reproduction of all images.

### Images

The schematic diagram in Fig 1 and stakeholder mapping diagram in Fig 2 was produced by the research team. Permissions have been secured from participants for the reproduction of all the images taken during the research. All images are owned by the VAPAR programme and reproduced in this paper under a creative commons license (Fig 5).

## Author Contributions

**Conceptualization:** Maria van der Merwe, Kathleen Kahn, Sophie Witter, Lucia D'Ambruoso.

**Data curation:** Jennifer Hove, Denny Mabetha, Maria van der Merwe, Rhian Twine, Kathleen Kahn, Sophie Witter, Lucia D'Ambruoso.

**Formal analysis:** Jennifer Hove, Denny Mabetha, Maria van der Merwe, Rhian Twine, Kathleen Kahn, Sophie Witter, Lucia D'Ambruoso.

**Funding acquisition:** Kathleen Kahn, Sophie Witter, Lucia D'Ambruoso.

**Investigation:** Jennifer Hove, Denny Mabetha, Maria van der Merwe, Rhian Twine, Kathleen Kahn, Sophie Witter, Lucia D'Ambruoso.

**Methodology:** Jennifer Hove, Denny Mabetha, Maria van der Merwe, Kathleen Kahn, Sophie Witter, Lucia D'Ambruoso.

**Project administration:** Denny Mabetha, Maria van der Merwe, Rhian Twine, Lucia D'Ambruoso.

**Resources:** Kathleen Kahn, Lucia D'Ambruoso.

**Supervision:** Rhian Twine, Kathleen Kahn, Lucia D'Ambruoso.

**Writing – original draft:** Jennifer Hove.

**Writing – review & editing:** Denny Mabetha, Maria van der Merwe, Rhian Twine, Kathleen Kahn, Sophie Witter, Lucia D'Ambruoso.

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
