## [Decision Letter · Decision Letter 0]

17 Mar 2023

PONE-D-22-28823

Participatory action research to address lack of safe water, a community stakeholder nominated priority in

PLOS ONE

Dear Dr. D'Ambruoso,

Thank you for submitting your manuscript to PLOS ONE. After careful consideration, we feel that it has merit but does not fully meet PLOS ONE’s publication criteria as it currently stands. Therefore, we invite you to submit a revised version of the manuscript that addresses the points raised during the review process.

This paper focused on an important subject. It is practical work and can be used by managers and policymakers to use participatory action research to address the lack of safe water.

The reviewers have raised a number of points that we believe would improve the manuscript. In addition to the items raised by the reviewer, please address the following points before more consideration:

It is better to state a specific study design in the first paragraph of the method section.

Which is the level of participation or the role of the research team? It must be presented in the method section.

The data analysis section should include which software were used. Did participants provide feedback on the findings?

Finally, the conclusion could be improved. What is the take-home message of this paper? Here, some conclusive remarks may be specified by the authors for the general readership.

We look forward to receiving your revised manuscript.

Kind regards,

Kamal Gholipour, PhD

Academic Editor

PLOS ONE

Journal Requirements:

Reviewers' comments:

Reviewer's Responses to Questions

**Comments to the Author**

1. Is the manuscript technically sound, and do the data support the conclusions?

Reviewer #1: Yes

Reviewer #2: No

Reviewer #3: Partly

2. Has the statistical analysis been performed appropriately and rigorously? 

Reviewer #1: Yes

Reviewer #2: No

Reviewer #3: N/A

3. Have the authors made all data underlying the findings in their manuscript fully available?

Reviewer #1: Yes

Reviewer #2: Yes

Reviewer #3: No

4. Is the manuscript presented in an intelligible fashion and written in standard English?

Reviewer #1: Yes

Reviewer #2: Yes

Reviewer #3: Yes

5. Review Comments to the Author

Reviewer #1: This paper provides as a case study from South Africa on community participatory research, and feedback on what interventions were ultimately possible and which did not succeed. The paper also provides a strong case for better listening to communities on what their problems are, and also realistic feedback that not all of these problems may be able to be solved as an outcome.

1. Additional background information is needed on why a lack of access to safe water remains in the study area. Is it due to rolling blackout of power which impacts the water pumping, or is there simply not enough water quantity in the area available? South African citizens are guaranteed water so how does this fit into the PAC?

2. Line 135 is unclear, why only one cycle and what are the other cycles?

3. Why did the community want a recycling center and beyond the presence of the NGO was there a community advocate?

4. Line 403 is missing a reflection on the election cycle impacting the mayor. Was there any one community issue that the mayor supported more or less?

5. The paper is missing a discussion on the strengths of this stakeholder engagement timeline, because other work shows that when it is too fast it fails. An additional figure with a timeline would assist the reader in understanding the many listening sessions and final outcomes.

6. Was there any reflection on the international authorship team here and the community listening sessions? Or, that partnership model?

7. In figure 1 and 2 please write out acronyms in full, it is hard to follow them.

8. The results and discussion are well written, but could be strengthen with more reflection on the “failures.” Can the authors give more details on outcomes not achieved and key points that can be pointed to when outcomes were achieved. Case studies such as this benefit literature the most in not overlooking both.

Reviewer #2: The article is not suitable for publication in this journal due to non-observance of the article writing format, for example, non-observance of standards in materials and methods and presentation of inappropriate tables and graphs.

Reviewer #3: Reviewer’s Comments

09/02/2023

Art. No.: PONE-D-22-28823

Art. Title: Participatory action research to address lack of safe water, a community stakeholder nominated priority in rural South Africa

Respectfully Professors,

Thank you for inviting me to review this document. The authors investigated the participatory action research to address lack of safe water in rural villages in South Africa.

The decision

I believe the breadth of this study is significant, and I highly suggest that the manuscript be published after minor adjustments. Thank you kindly.

Major issues: -

1- The manuscript should be checked carefully for English and formatting style.

2- The authors should provide Figure no. 3?

3- The authors should reorganize the methods in simple way?

4- The authors should provide flowchart summarize the study design?

Thanks

6. PLOS authors have the option to publish the peer review history of their article (what does this mean?). If published, this will include your full peer review and any attached files.

Reviewer #1: No

Reviewer #2: No

Reviewer #3: No

---

## [Author Response · Author response to Decision Letter 0]

13 May 2023

Reviewer(s)' Comments to Author(s): 

Reviewer #1: 

This paper provides as a case study from South Africa on community participatory research, and feedback on what interventions were ultimately possible and which did not succeed. The paper also provides a strong case for better listening to communities on what their problems are, and also realistic feedback that not all of these problems may be able to be solved as an outcome Many thanks for your careful review and insightful comments. We have carefully addressed all the comments and suggestions and we believe that the revised version is now suitable for publication in PLOS ONE. We provide responses below 

1. Additional background information is needed on why a lack of access to safe water remains in the study area. Is it due to rolling blackout of power which impacts the water pumping, or is there simply not enough water quantity in the area available? South African citizens are guaranteed water so how does this fit into the PAC? The resubmitted manuscript now includes additional background information on the persistent challenge of water access in the Bushbuckridge local municipality study area. While recent power cuts may be a contributing factor, they are not the primary cause of the water access issue in the region. Data collection for cycle one was conducted from 2017 -19, during which the situation was challenging, but it was no acute as it is in 2023. (Page 5)

2. Line 135 is unclear, why only one cycle and what are the other cycles? In this paper we report on the first cycle which comprised three steps described below. These steps were crucial in informing the subsequent steps reported elsewhere. (Page 8)

3. Why did the community want a recycling centre and beyond the presence of the NGO was there a community advocate? These action items were developed in collaboration with local communities, NGOs, and researchers. Specifically, the NGO led the implementation of this particular action item in partnership with local communities. (Page 17)

4. Line 403 is missing a reflection on the election cycle impacting the mayor. Was there any one community issue that the mayor supported more or less? During the planning phase, it is crucial to time the project in a way that ensures the mayor's availability for support, buy-in, and participation, without other commitments. This is particularly important during elections when the mayor's availability may be limited. (Page 23)

5. The paper is missing a discussion on the strengths of this stakeholder engagement timeline, because other work shows that when it is too fast it fails. An additional figure with a timeline would assist the reader in understanding the many listening sessions and final outcomes. Agreed, and we have added more discussion on the strengths of stakeholder engagement timeline in the discussion section and a figure. However, this paper is reporting on the 1st of 5 cycles of the wider programme of research.(Page 28)

6. Was there any reflection on the international authorship team here and the community listening sessions? Or, that partnership model? Thank you for these constructive suggestions. We have tried to address this by adding a researcher reflection in the result section.(Page 28)

7. In figure 1 and 2 please write out acronyms in full, it is hard to follow them Thank you, this has been corrected as suggested 

8. The results and discussion are well written but could be strengthen with more reflection on the “failures.” Can the authors give more details on outcomes not achieved and key points that can be pointed to when outcomes were achieved. Case studies such as this benefit literature the most in not overlooking both. The authors thank the reviewer for urging us to incorporate these contextual features. The result and discussion section of the revised manuscript gives more details on the outcomes not achieved. 

Reviewer #3 

I believe the breadth of this study is significant, and I highly suggest that the manuscript be published after minor adjustments. Thank you kindly.

 Thank you for the encouraging comments and positive feedback, which has helped us to improve the quality of our manuscript. 

1. The manuscript should be checked carefully for English and formatting style We have formatted and reviewed the manuscript.

2. The authors should provide Figure no. 3? Thank you. Figure 3 provided 

3. The authors should reorganize the methods in simple way? We have reviewed the methods section and added stakeholder timeline diagram.

4. The authors should provide flowchart summarize the study design? Thank you for this suggestion, we have added a schematic illustration of the methods. 

END OF RESPONSE TO REVIEW

---

## [Decision Letter · Decision Letter 1]

29 Jun 2023

Participatory action research to address lack of safe water, a community-nominated health priority in rural South Africa

PONE-D-22-28823R1

Dear Dr. D'Ambruoso,

We’re pleased to inform you that your manuscript has been judged scientifically suitable for publication and will be formally accepted for publication once it meets all outstanding technical requirements.

Kind regards,

Kamal Gholipour, PhD

Academic Editor

PLOS ONE

Additional Editor Comments (optional):

Reviewers' comments:

Reviewer's Responses to Questions

**Comments to the Author**

1. If the authors have adequately addressed your comments raised in a previous round of review and you feel that this manuscript is now acceptable for publication, you may indicate that here to bypass the “Comments to the Author” section, enter your conflict of interest statement in the “Confidential to Editor” section, and submit your "Accept" recommendation.

Reviewer #1: (No Response)

2. Is the manuscript technically sound, and do the data support the conclusions?

Reviewer #1: Yes

3. Has the statistical analysis been performed appropriately and rigorously? 

Reviewer #1: Yes

4. Have the authors made all data underlying the findings in their manuscript fully available?

Reviewer #1: No

5. Is the manuscript presented in an intelligible fashion and written in standard English?

Reviewer #1: No

6. Review Comments to the Author

Reviewer #1: Please revisit the following 3 comments which have not been addressed satisfactory from the first round.

3.Why did the community want a recycling center and beyond the presence of the NGO was there a community advocate?

4.Line 403 is missing a reflection on the election cycle impacting the mayor. Was there any one community issue that the mayor supported more or less?

5.The paper is missing a discussion on the strengths of this stakeholder engagement timeline, because other work shows that when it is too fast it fails. An additional figure with a timeline would assist the reader in understanding the many listening sessions and final outcomes.

Finally, when you submit the corrected version, please do check thoroughly, in order to avoid grammar, syntax or structure/presentation flaws - please seek for professional English proofreading services or ask a native English-speaking colleague of yours in order to refine and improve the English in your paper.

7. PLOS authors have the option to publish the peer review history of their article (what does this mean?). If published, this will include your full peer review and any attached files.

Reviewer #1: No

---

## [Editor Report · Acceptance letter]

19 Jul 2023

PONE-D-22-28823R1 

Participatory action research to address lack of safe water, a community-nominated health priority in rural South Africa 

Dear Dr. D'Ambruoso:

I'm pleased to inform you that your manuscript has been deemed suitable for publication in PLOS ONE. Congratulations! Your manuscript is now with our production department. 

Kind regards, 

on behalf of

Dr. Kamal Gholipour 

Academic Editor

PLOS ONE